# Environment Friendly Pretreatment Approaches for the Bioconversion of Lignocellulosic Biomass into Biofuels and Value-Added Products

Surbhi Sharma [1,†], Mei-Ling Tsai [2,†], Vishal Sharma [2,3,†] , Pei-Pei Sun [2], Parushi Nargotra [2], Bijender Kumar Bajaj [1], Chiu-Wen Chen [3,4,*] and Cheng-Di Dong [3,*]

1 School of Biotechnology, University of Jammu, Jammu 180006, India
2 Department of Seafood Science, National Kaohsiung University of Science and Technology, Kaohsiung City 81157, Taiwan
3 Department of Marine Environmental Engineering, National Kaohsiung University of Science and Technology, Kaohsiung City 81157, Taiwan
4 Institute of Aquatic Science and Technology, National Kaohsiung University of Science and Technology, Kaohsiung City 81157, Taiwan
* Correspondence: cwchen@nkust.edu.tw (C.-W.C.); cddong@nkust.edu.tw (C.-D.D.)
† These authors contributed equally to this work.

**Abstract:** An upsurge in global population and rapid urbanization has accelerated huge dependence on petroleum-derived fuels and consequent environmental concerns owing to greenhouse gas emissions in the atmosphere. An integrated biorefinery uses lignocellulosic feedstock as raw material for the production of renewable biofuels, and other fine chemicals. The sustainable bio-economy and the biorefinery industry would benefit greatly from the effective use of lignocellulosic biomass obtained from agricultural feedstocks to replace petrochemical products. Lignin, cellulose, hemicellulose, and other extractives, which are essential components of lignocellulosic biomass, must be separated or upgraded into useful forms in order to fully realize the potential of biorefinery. The development of low-cost and green pretreatment technologies with effective biomass deconstruction potential is imperative for an efficient bioprocess. The abundance of microorganisms along with their continuous production of various degradative enzymes makes them suited for the environmentally friendly bioconversion of agro-industrial wastes into viable bioproducts. The present review highlights the concept of biorefinery, lignocellulosic biomass, and its valorization by green pretreatment strategies into biofuels and other biochemicals. The major barriers and challenges in bioconversion technologies, environmental sustainability of the bioproducts, and promising solutions to alleviate those bottlenecks are also summarized.

**Keywords:** integrated biorefinery; circular bioeconomy; lignocellulosic biomass; pretreatment; valorization; enzymes; biofuels; biochemicals

## 1. Introduction

Global population growth and socioeconomic expansion have contributed to an enormous rise in pollution and an expedited depletion of energy resources [1–3]. The primary cause of the current global energy crisis and deterioration of the environment is the over-consumption of conventional fossil fuels, which contribute significantly to greenhouse gas emissions, global warming, acid rain, and other climatic ill-effects [4,5]. Thus, it is imperative to make enduring technological developments to supply the accelerating energy demand without imperiling the planet's finite resources [6]. The paradigm of 'take, make, and dispose of' is currently being replaced by 'reuse and recovery' of resources in order to attain 'healthy climate-healthy individuals' and socioeconomic success [7]. To produce low-carbon, long-lasting, and fossil-free fuels, the development of various

non-conventional/renewable/infinite energy resources is being explored. Lignocellulosic biomass (LCB)-based biofuels are attaining prominence as a feasible substitute for petroleum-based fuels [4,8]. LCB constituted by agricultural residues/food waste/forest products/municipal and sewage wastes represents the most promising, copious, and a low-cost carbon-based substrate, which can be extensively exploited for the production of biofuels and other commercial products in a biorefinery [5,9–11]. The sustainable bioeconomy and the biorefinery industry would benefit greatly from the effective use of lignocellulosic biomass obtained from agricultural feedstocks to replace petrochemical products.

Biorefinery is a strategy which integrates several conversion processes, such as biochemical, thermo-chemical, and microbial growth to generate energy, biofuels, chemicals, and other value-added desired industrial products from LCB [12]. However, the full-scale bioconversion of LCB to bioenergy/biofuels (biodiesel, bioethanol, biobutanol), and biobased commodities (chemicals, feed, food) poses various significant obstacles in terms of energy input and product yield in the overall process [13]. LCB, a hetero-polysaccharide complex, is composed of a variety of polymers (cellulose, hemicellulose, and lignin), and other polar and non-polar substances [10,14–16]. The structural polysaccharides of LCB are entangled with lignin blocks in a very complicated manner, thereby imparting high rigidity and recalcitrance to the biomass structure [13]. Hence, a key bottleneck in the valorization of LCB for the production of power, fuels, or other chemicals is its structural recalcitrance, which might be successfully reduced by implementing an appropriate pretreatment approach [1,17]. Pretreatment aims to disrupt the crystalline structure of biomass by breaking the hydrogen bonds and cross-linked hydrophobic interactions between cellulose, hemicellulose, and lignin; thus, increasing the accessibility of biomass to enzyme hydrolysis [18,19]. An effective pretreatment may improve the cellulose's accessibility, pore size, and surface area while lowering its degree of polymerization (DP) and crystallinity [20,21]. The primary pretreatment regimes available for the fractionation of LCB can be categorized as physical, chemical, physicochemical, and biological techniques [7]. However, most of these methods possess several limitations such as high cost, being non-'green', generation of inhibitors/toxic substances, and many others. Therefore, in the current times, research is being focused on the development of environmentally friendly and cost-efficient pretreatment methods to achieve successful LCB fuel technology [14,16]. The choice of an appropriate pretreatment technique depends upon biomass type, its structural properties, composition, and degree of polymerization. Pretreatment is the principal capital-intensive step in LCB biorefinery; thus, it significantly affects the overall economy of a bioprocess [22].

Another crucial step in the chain of LCB fuel schemes is enzymatic saccharification of suitably pretreated biomass. The enzyme cocktail of cellulases (endo-β-1,4-glucanase, cellobiohydrolase, and β-glucosidase), and hemicellulases (xylanases, glucuronidase, glucomannanase, galactomannanase, β-xylosidase, acetylesterase) works synergistically to rupture the cellulose and hemicellulose fibers into glucose monomers during enzymatic hydrolysis [20,23]. Currently, lytic polysaccharide monooxygenases (LPMOs), manganese peroxidase, and laccases are being employed for enzymatic saccharification because they hydrolyze the polysaccharides and remove lignin; thus, improving the effectiveness of process by increasing the glucose yield and decreasing the quantity of cellulases [4,23]. However, high sensitivity, high cost, less stability, and high dosage of the enzymes hinder greatly the economic feasibility of the biofuel production process. Thus, novel approaches such as onsite enzyme production, usage of enzyme cocktails, enzyme immobilization, and others are being utilized for achieving the cost-economy of the overall process [14,24]. Besides biofuel, a variety of value-added products are also produced through enzymatic saccharification of pretreated LCB, such as xylo-oligosaccharides, xylose, psicose, tagatose, xylene, toluene, benzene, aliphatic acids, quinines, and lignin monomers including syringols, syringaldehyde, propylphenol eugenol, vanillin, aryl ethers, and many others [25]. For the production of biofuel–ethanol, butanol, lactic acid, acetone, or other end products, the reducing sugars (hexoses and pentoses) obtained are subjected to a biochemical pro-

cess named fermentation [23]. The fermentation process is intended to be optimized via pretreatment and hydrolysis techniques [26].

Based on the composition of sugary hydrolysate, specific micro-organisms (bacteria, fungi or yeasts) are required for metabolization and product formation [2]. The commonly used micro-organisms for ethanol fermentation include *Saccharomyces cerevisiae*, *Pichia stipitis*, *Zymomonas mobilis*, *Candida shehatae*, *Candida brassicae*, *Klebsiella oxytoca*, *Mucor indicus*, *Escherichia coli*, *Penicillium*, *Trichoderma*, and *Aspergillus* [9,15,18,27]. *Saccharomyces cerevisiae* is the principal yeast which has been employed extensively for the production of alcohol, mainly in wine industries and breweries [2]. Thus, valorization of lignocellulosic feedstock into biofuels, and other upgraded products, envisages the 'waste to wealth' transformation, and the growth of a biomass-based economy [28]. The current review therefore addresses the concept of biorefinery, the associated challenges, and the possible solutions that might improve bioprocessing for the development of a sustainable biorefinery. The recent trends in the different steps of bioconversion of LCB into valuable bioproducts, including pretreatment and enzymatic hydrolysis steps, are presented. Additionally, a brief description of promising physical, chemical, physicochemical, and biological pretreatment strategies and key steps involved in biofuel production processes is outlined. The review also discusses the environment sustainability of the biofuels and bioproducts, current challenges, and possible sustainable solutions to achieve the targets of integrated biorefineries.

## 2. Lignocellulosic Biorefinery Concept for Circular Bioeconomy

In order to boost production of green fuels and add economic value, the concept of a biorefinery was established. A biorefinery is a fundamental framework developed for the use of raw biomass (LCB, algae, other organic wastes), in which all processing methods are integrated in a systematic manner to produce sustainable bio-based products [2]. The biorefineries are essential to foster a circular bioeconomy that is knowledge-driven and environmentally sound [2,29].The idea of circular bioeconomy emphasizes the recovery of all goods from resources without any waste being discharged, and the lignocellulosic biomass utilized as a feedstock plays a vital role in the production of both bioproducts and the energy needed to power them. The primary goal of circular bioeconomy is to replace the end-of-life phase through resource recovery, the use of renewable energy, the elimination of harmful chemicals, and the zero discharge concept by altering the design of all relevant systems and business models [12]. Conventional biorefineries implement high levels of process integration and little waste output to transform biomass into a variety of products for different industrial sectors [30]. According to the International Energy Agency (IEA), biorefinery targets "the sustainable conversion of various biomasses into bioenergy and diverse bioproducts (chemicals, feed, and food), and bioenergy (power, heat, and biofuels)" [26]. Based on the literature survey [2,20,31], biorefinery can be classified into three phases on the basis of biomass, biomass processing technology, and the product:

a.　Phase I biorefinery (one feed, one preset processing technology, and one main product). For example, in the European Union, vegetable oil biorefinery involves the production of biodiesel by the transesterification of rapeseed oil;

b.　Phase II biorefinery (one feed, multiple processing technologies, and multiple end products). For example, in Sweden, forestry feedstock is refined to produce cellulose, lignosulphonate, and bioethanol;

c.　Phase III biorefinery (multiple feedstocks, multiple processing technologies, and multiple end products). For example, lignocellulosic biomass-based biorefinery, two-platform biorefinery, green biorefinery, and others.

A successful biorefinery must be the one which produces very little or no waste, while processing. Thus, an integrated strategy which combines conversion strategies (chemical, thermochemical, and biochemical), and potent downstream strategies systematically, must be used in a zero-waste biorefinery for achieving the target of circular bioeconomy [2,12]. In this way, a sustainable and more economical approach could be developed for an ideal biorefinery, which could potentially provide a plethora of fuels, and value-added industrial

products [23]. With the basic layout of a biorefinery process, the unexplored lignocellulosic biomass can serve as a potent resource for the production of bioproducts. The ability to use various LCBs can provide a consistent supply for biorefining processes throughout the year; thus, boosting the viability of biomass waste valorization with an economic perspective [32]. The accomplishment of LCB-based second generation (2G) biorefinery is largely determined by the type, composition, and properties of biomasses as well as the conversion processes used in the process.

## 3. Lignocellulose Composition and Structure

Lignocellulosic biomass is the most copious, low-cost, carbon-neutral, and high-energy density organic polymer present on this earth [33]. The production of renewable fuels from LCB not only boosts global energy security, but also minimizes waste, helps rural economies, and reduces environmental problems [34]. LCB is a heterogeneous polymeric complex composed of three major components; i.e., cellulose (35–55%), hemicellulose (20–40%), and aromatic lignin (10–25%), and minor traces of protein, pectin, ash, and other extractives (inorganics, waxes, fats, resin acids, phenolics) [12,35]. Table 1 shows the cellulose, hemicellulose, and lignin content in different types of lignocellulosic feedstocks.

**Table 1.** Composition of different lignocellulosic biomass feedstocks.

| S. No. | Lignocellulose Feedstocks | Cellulose (%) | Hemicelluloses (%) | Lignin (%) | Reference |
|---|---|---|---|---|---|
| 1. | Sugarcane bagasse | 38.6 | 20 | 24.68 | [9] |
| 2. | Sunflower stalks | 27.22 | 11.94 | - | [35] |
| 3. | Oat flakes | 21 | 38% | 27 | [36] |
| 4. | Spruce sawdust | 55.4 | 1.4% arabinose, 4.2% xylose | 28.7 | [37] |
| 5. | Eucalyptus | 41.58 | 15.85 | 29.40 | [38] |
| 6. | *Parthenium hysterophorus* | 49.98 | 7.61%arabinose, 14.18% xylose | 17.6 | [15] |
| 7. | *Saccharum spontaneum* | 32.16 | 19.36 | 16.86 | [33] |
| 8. | Birchwood planks | 54.22 | 28.14 | 11.13 | [39] |
| 9. | Oak sawdust | 44.7 | 1.2% arabinose, 14.8% xylose | 26.7 | [37] |
| 10. | Oil palm trunk | 56.1 | 16.15 | 19.11 | [40] |
| 11. | Pine | 36.2 | 23.0 | 32.8 | [41] |
| 12. | Cup plant | 39 | 21 | 21 | [42] |
| 13. | Sun hemp fiber | 75.6 | 10.05 | 10.32 | [43] |
| 14. | Watermelon rind | 39.67 | 23.21 | 10.6 | [44] |
| 15. | Peanut shell | 36.9% glucan | 13.2% xylan, 1.5% galactan, 5.2% arabinan, 1.0% mannan, | 30.2% Klason lignin, 3.9% acid-soluble lignin, | [45] |
| 16. | Corn cob | 41 | 22.6 | 14.1 | [46] |
| 17. | Barley straw | $31.1 \pm 0.8$ | $27.2 \pm 0.4$ | $18.8 \pm 0.2$ | [47] |
| 18. | Corn stover | 31.5% glucan | 22.5% xylan, 2.1% galactan, 1.7% arabinan | 18 | [48] |

The percentage composition of these polymers may differ in different biomasses due to variations in biomass type (softwood, hardwood, grasses, and farm waste), soil type, climate type, fertilizers type, growing conditions, harvesting technique, and other physical factors [1,9,16].

### 3.1. Cellulose

Cellulose ($C_6H_{10}O_5$)$_x$ is an unbranched linear polymer of β-D-glucopyranose units, joined together by β-(1,4) glycosidic linkages [49]. It is the primary component of lignocellulosic biomass, which accounts for 30–50 wt% of the whole dry mass [2]. The degree of polymerization (DP) of cellulose fibrils is approximately greater than 10,000. The uniform

structure of cellulose fibrils is formed by inter- and intramolecular hydrogen bonding as well as covalent interactions, which offers great tensile strength to cellulose and makes it resistant to various solvents [34]. Due to the presence of hydrogen bonding between hydroxyl groups in cellulose, different crystalline patterns with varying degrees of crystallinity are produced.

The cellulose polymers are separated from one another and more likely to form hydrogen bonds with other molecules in the more amorphous regions. Due to these two distinct molecular configurations, the polymer has a paracrystalline structure, where both the amorphous and crystalline portions are present [50]. Amorphous cellulose, which is easier to access, can be easily hydrolyzed by cellulase, while the harder-to-access crystalline part cannot be effectively broken down [36]. The important bio-based products obtained from the processing of cellulose include ethanol, succinic acid (SA), levulinic acid (LA), sorbitol, hydroxyacetaldehyde, and many others [2].

### 3.2. Hemicellulose

Hemicellulose $(C_5H_8O_4)_m$, the most rich carbohydrate polymer after cellulose, constitutes about 15–35 wt% (dry mass basis) of total biomass. It is structurally a heteropolymer with less polymerization degree and molecular weight as compared to cellulose [23,27]. It contains both branching and linear polymers that are made from a variety of anhydrous sugars (xylose, arabinose, D-glucuronic acid, L-rhamnose, and others). Since the hemicellulose components have amorphous nature, and low DP, they do not significantly increase the biomass's recalcitrance [1,49]. Thus, it can be separated rapidly, and utilized to make biofuels such as bioethanol, and other value-added products including lactic acid, xylo-oligosaccharides, xylitol, hydroxymethylfurfural (HMF), polyhydroxyalkanoates, and furfural [2]. The most prevalent hemicellulose polysaccharides are known as xylans, which have polymeric chains made up of 1,4-linked -D-xylose units [51]. Depending on its source, xylan has a different chemical makeup. Hemicellulose also contains mannan components such as glucomannan, galactomannan, glucuronic acid, and galacturonic acid. Mannose residues are connected by β-1,4-bonds, whereas galactomannan is formed when mannose residues use α-1,6-bonds to link the galactose residues.

### 3.3. Lignin

Lignin $[C_9H_{10}O_3(OCH_3)_{0.9-1.7}]_n$ is primarily one of the crucial components of plant cell walls that offers them stiffness and strength while guarding against microbial attack [1,52]. It is an insoluble, stiff, aromatic biopolymer made up of monolignols such assinapyl alcohol, p-coumaryl alcohol, and coniferyl alcohol [20,53]. The presence of these components varies significantly, depending on the type of plant and extraction technique, with softwood and hardwood having higher (25–31 wt%) and lower percentages (16–24 wt%) of lignin, respectively [15,17,35]. With a projected accessibility of more than $3 \times 10^{11}$ tons, or around 30% of all non-fossil carbon, it is a rich organic non-fossil carbon present on the planet. Due to its abundant availability, lignin macromolecule represents a potential source for the production of fine chemicals, adsorbents, fuels, and polymer processing materials [54]. Lignin generally forms a three-dimensional cross-linked network by integrating with cellulose and hemicellulose components of LCB.

The common interunit bonds in lignin include β-aryl ether resinol, phenylcoumaran, dibenzodioxocin, β-1,4-O-5′, and α-O-4′ linkages [53]. Ether bonds, which make up roughly 60–75% of all bonding in lignin and are a common type of bond accounting for approximately 45–62% of all connection types, are the main type of bond used to connect the various structural units [1]. Lignin has long been considered a roadblock in the lignocellulosic biorefineries due to its non-productive binding with the cellulolytic enzymes [3,55]. However, the past ten years have brought a paradigm shift in lignin research so that it is now considered a bridge for biorefineries and an essential natural polymer for value addition that can achieve the targets of circular bioeconomy in future.owe

## 3.4. Other Components

Apart from cellulose, hemicellulose, and lignin polymers, LCB is also constituted by proteins, pectins, small amounts of extractives (5–15 wt%) such as gums, terpenoids, diterpenes, fatty acids, and chlorophyll, other phenolic substances, and mineral-rich ash (1–5 wt%) (K, Mg, Ca, Si) [34,51]. In contrast to lignocellulosic content, these components provide color and smell to the wood, serve as energy storage units, and shield the plant species from microbial invasions [22]. The non-structural parts of lignocellulosic biomass, known as extractives, can be dissolved in water or neutral organic solvents such as hexane and ethanol. Various biopolymers, including terpenes, terpenoids, proteins, fats, waxes, lipids, steroids, and phenolic substances including lignans, stilbenes, and tannins, are found in extractives [51]. Ash is a solid, inorganic residue that remains after all LCB deposits have been completely burned [56]. It contains elements and minerals such as silicates, carbonates, phosphates, calcium, magnesium, sodium, and potassium [57]. In contrast to agricultural wastes, woody biomass often has a very low ash content [51]. For the prediction of deposits that form in the boilers during gasification or incineration, it is crucial to have a thorough understanding of the physical and chemical features of lignocellulosic ashes.

## 4. Recent Advances in Pretreatment Technologies for Lignocellulosic Biomass

The highly complex and recalcitrant structure of lignocellulosic biomass is the biggest barrier to 2G biorefining [58]. The intermeshed matrix of LCB biopolymers (cellulose, hemicellulose, and lignin) provides resistance to various microbial and enzymatic digestions. The other variables that affect the biomass recalcitrance include the biomass's porosity, and the presence of proteins and acetyl groups [59]. Thus, fractionation of lignocellulosic components becomes essential for the extraction of monomeric sugars to produce biopolymers, biofuels, and biochemicals [1,4,33]. Pretreatment of LCB is a very significant step in a biorefinery, which enables the efficient disintegration of biomass, increasing the permeability and surface area availability of polysaccharides towards enzymatic hydrolysis. A desirable pretreatment aims to reduce biomass crystallinity, improve porosity, remove lignin, prevent inhibitor formation, avoid sugar degradation, require less energy input, and cost less [36,60]. A variety of conventional pretreatment strategies have been developed for the disruption of biomass, such as biological, physical, chemical, and physicochemical (Figure 1) [1,51]. Table 2 shows the merits and limitations of different pretreatment methods along with their effects in the valorization of LCB. The majority of these methods, however, are associated with limitations such as high operational expense, harsh operational conditions (pressure, pH, and temperature), toxic product formation, and many others [14,15,61]. Hence, novel eco-benign pretreatment techniques are needed to be developed to overcome the constraints of traditional methods.

### 4.1. Physical Pretreatment

Physical pretreatment techniques primarily aim to decrease the particle size of LCB, and increase the surface accessibility of polysaccharides [22,62]. Mechanical treatment (chipping, milling, and/or grinding), freezing, pyrolysis, microwave irradiation, pulsed electric field, ultrasonication, and torrefaction are several pretreatment approaches employed to destruct the lignocellulosic biomass [49,62,63]. However, most of these methods necessitate high energy/power consumption, high-cost processing instruments/equipment, and also produce hazardous chemical compounds. Thus, they are considered less attractive for industrial-scale usage [5,22,34,61]. Even so, various methods including ultrasound and microwave irradiation have been developed as very efficient in the decrystallization and solubilization of cellulose and lignin from lignocellulosic feedstock.

### 4.1.1. Ultrasound Pretreatment

Ultrasound irradiation assisted pretreatment includes application of high-frequency (20–500 MHz) mechanical acoustic waves that produce a high-shear force for proficient defibrillation, and degradation of lignocellulosic biomass [18]. Ultrasound pretreatment

method is based on the basic principle of the 'cavitation effect', in which the ultrasonic waves aid in breaking α-O-4 and β-O-4 lignin linkages, which results in the development of cavitation bubbles by fractionating the ligno-polysaccharides complex of biomass [7,9]. When these bubbles reach a threshold size, they rapidly collapse, causing a significant rise in temperature (2000–5000 K) and pressure (1800 atm) [23]. This also offers a number of benefits including less residence time, high levels of activation energy, and adequate mass transfer for efficient LCB deconstruction [62].

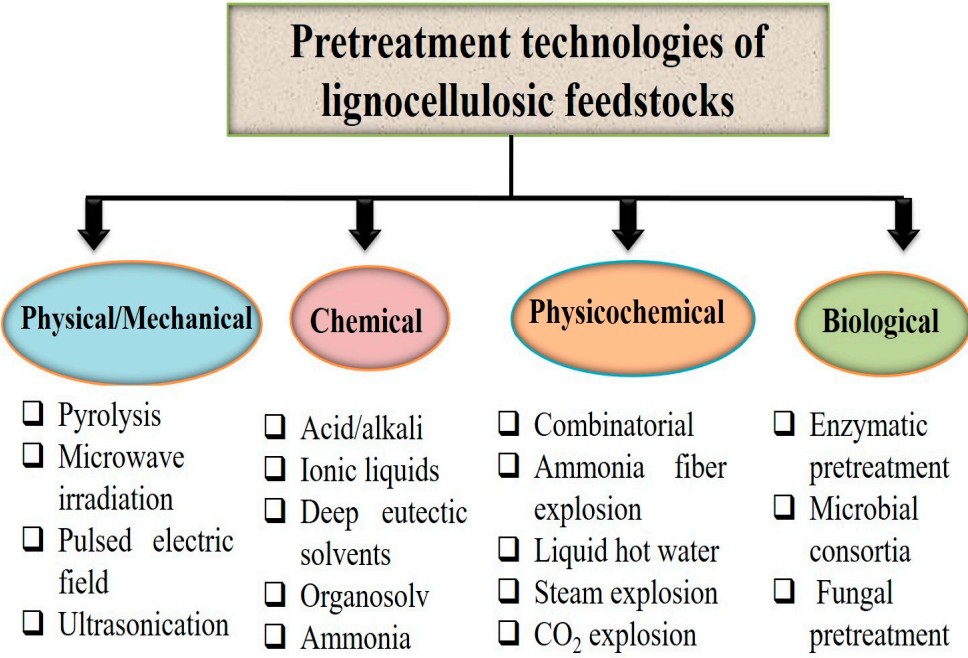

**Figure 1.** Different pretreatment technologies for breaking the inherent recalcitrant structure of lignocellulosic biomass.

He et al. [64] investigated the structural variations of eucalyptus wood due to ultrasound pretreatment assisted with distilled water, aqueous acetic acid, or aqueous soda solution at 28 KHz frequency and 300 W power. It was observed that the proportion of alkali metals was significantly reduced after ultrasound pretreatment. There was an increase in the crystallinity (35.5%) of wood samples after pretreatment, which indicated the successful disintegration of wood biomass. Subhedar et al. [65] reported pretreatment of pistachio shells, groundnut shells, and coconut coir using ultrasound irradiation (0.5% *w/v* biomass loading, 100W power, and 70 min pretreatment time). Compared to alkaline pretreatment, the delignification percentage of LCB was significantly higher (80–100%) in case of the ultrasound-assisted method. At optimal parameters, the sugar yield obtained through ultrasound-assisted digestion was found to be greater for pistachio shells, groundnut shells, and coconut coir, i.e., 18.4, 21.3, and 23.9 g/L, respectively, as compared to alkali hydrolysis, i.e., 8.1, 10.2, and 12.1 g/L, respectively, in similar biomass order. In another study conducted by Kunaver et al. [66], a high-frequency ultrasound irradiation approach was employed for the liquefaction of different forest wood waste residues into value-added products. The results demonstrated the high production of low-molecular-weight residual products by using short reaction time (decreased by factor 9), and less energy/power consumption. Thus, ultrasound technology (standalone/integrated) has been the top choice for effective LCB pretreatment.

### 4.1.2. Microwave Irradiation

This irradiation method involves non-ionizable, low-frequency electromagnetic radiation waves called microwaves produced by cycling off a dipole (two opposite and equal charges separated by a particular distance) [22]. Microwave pretreatment exhibits an

excellent potential in the bioprocessing of lignocellulosic biomass as the heat produced by the technique raises the temperature and pressure (thermal effect), throughout the biomass rather than just on its surface, demonstrating greater energy efficiency than traditional heating methods [67]. According to Mankar et al. [62], LCB acts as a good biological conductor due to the presence of non-ionic amorphous and ionic crystalline cellulose regions. Temperature, initial feedstock components, irradiation time, microwave power, and catalyst loading have a discernible impact on the microwave–physicochemical processing of biomass [68]. Application of microwave irradiation heats the polar structure of LCB, and leads to the formation of hotspots. Consequently, constant blasts occur inside the biomass structure, which accelerates the reorientation of crystalline cellulose fibrils [67]. In particular, microwave heating has a number of benefits, including short reaction time, fast and homogenous heating, low activation energy, increased product yield, few by-products, being environmentally friendly and a low-cost, energy-effective operation [68].

Nuchdang et al. [69] investigated the effect of microwave pretreatment for *Brachiara mutica* (paragrass) biomass, treated with acid and alkali. The results showed that use of microwave irradiation at 120 °C, 30 min, and 5% $w/w$ alkali-to-biomass ratio increased the total reducing sugars significantly (137.3%), as compared to native biomass. Kainthola et al. [70] reported the highest solubilization of rice straw biomass under the influence of microwave irradiation (for 4 min, at 190 °C), resulting in a high specific yield of methane, i.e., 325.2318 mL/g/VS. A short reaction time and high heating rate reduced the production of inhibitory products, thus improving the efficacy of the pretreatment process. In another study, microwave-assisted $FeCl_3$, $H_2SO_4$, and NaOH pretreatment was carried out for sugarcane bagasse biomass to extract the fermentable sugars. The results demonstrated high lignin removal, and high sugar yield including removal of glucose and xylose selectively, at a short residence time of 5 to 10 min [71]. Despite the amazing advantages of microwave irradiation, the development of industrial microwave heating applications and the incorporation of microwaves into the physicochemical approach for biomass preparation are relatively limited. Additionally, the operation conditions for microwave-assisted pretreatment should be fine-tuned to extract the polymeric sugars (cellulose/hemicellulose) as efficiently as possible.

*4.2. Chemical Pretreatment*

This type of pretreatment involves the usage of various chemicals such as acids, alkali, deep eutectic solvents, ionic liquids, ammonia, ozone, and many others for the disruption of recalcitrant lignocellulosic biomass [1,15,33]. Chemical pretreatment aims to decrease the biomass crystallinity, increase its porosity, remove lignin, and thus, enhance the biomass surface availability towards enzymatic hydrolysis [60]. Acids (concentrated and dilute) cleave the hemicellulose fraction, and remove part of the lignin of LCB, thus providing greater cellulose access to hydrolysis by enzymes [72]. However, most acids are associated with various limitations such as sugar loss, high cost, corrosive nature, and production of inhibitors [23]. Morais et al. [73] carried out pretreatment of sugarcane bagasse with dilute phosphoric acid that under optimal pretreatment conditions (4.95% phosphoric acid, 80 °C, 375 min) yielded maximum reducing sugars (98% glucose) with higher saccharification efficiency (99%). Alkaline pretreatment leads to the delignification of biomass by cleaving aryl-ether, ester, and C-C bonds, removes hemicelluloses, and increases the surface area for efficient hydrolysis and fermentation [22,74]. However, long reaction time, and high expense for neutralization of pretreatment slurry are the major limitations of the alkaline pretreatment [62]. A mild alkali (sodium hydroxide)-mediated pretreatment of cogon grass was performed at 2% $w/v$ NaOH concentration, 90 min time, and 85 °C [75]. The results showed higher saccharification efficiency (90.8%) after alkali pretreatment, which on fermentation yielded maximum ethanol (76%). Although traditional chemicals (acids/alkalis) result in improved decrystallization and cellulose accessibility in a biorefinery, they also have a number of drawbacks that prevent their usage on a large scale [76]. These pretreatment techniques' principal downsides include the high expense of recovering

the acid/alkali, the requirement for corrosive-resistant equipment, and the production of inhibitory compounds. Therefore, the pursuit of new eco-friendly solvents has been intensified in recent years.

### 4.2.1. Ionic Liquids (ILs)

Ionic liquids (ILs) have effectively become the most environmentally friendly and reusable organic solvents for LCB processing due to their numerous extraordinary tuneable characteristics [13,18,53]. ILs are organic salts of cationic and anionic species, have lower melting point, are non-inflammable, have high chemical and thermal stability, and low vapor pressure [59]. IL-mediated pretreatment solubilizes the polar and non-polar molecules of ligno-polysaccharides, decreases cellulose crystallinity, increases porosity, eliminates the lignin sheath, and thus provides larger cellulose accessibility to saccharification enzymes. Several imidazolium-based ionic liquids have been used for pretreatment of lignocellulosic biomass.

Alayoubi et al. [37] carried out low-temperature (45 °C) IL, 1-ethyl-3-methylimidazolium acetate pretreatment of oak sawdust, spruce sawdust, and model cellulose, which on saccharification produced a high glucose yield, i.e., 59.3%, 49.3%, and 68.2%, respectively. Dotsenko et al. [77] reported an effective delignification, and biotransformation of hardwood (hornbeam wood and spruce), and softwood biomass samples, after pretreatment with ionic liquid 1-butyl-3-methylimidazolium chloride ([Bmim]Cl). Pretreatment of sugarcane tops biomass using ionic liquid tris (2-hydroxyethyl) methylammonium-methylsulfate ([TMA]MeSO$_4$), followed by saccharification, resulted in a high sugar yield, i.e., 181.18 mg/g biomass [14]. However, expensive synthesis, potential toxicity, recycling issues, and non-biodegradability are a few limitations of ILs which limit their large-scale usage in industries [9,78].

### 4.2.2. Deep Eutectic Solvents (DESs)

Lately, DESs have gained a lot of interest as a green chemistry approach for the digestibility of lignocellulosic biomass [1,9]. Deep eutectic solvents primarily consist of a mixture of liquid eutectic materials that include hydrogen-bond donors and acceptors at a particular molar ratio [79]. They exist mostly in liquid form, and possess lower melting points than their single constituents [16]. They exhibit almost similar characteristics with ILs including low vapor pressure, non-flammability, high thermo-chemical stability, high miscibility, high polarity, and thus, can be efficiently used for the fractionation of LCB [62,79]. However, easy synthesis, low cost, non-toxicity, good recyclability, and increased biodegradability make them preferable solvents to ILs. DESs solubilize lignin effectively from LCB, and transform the crystalline portion of cellulose into amorphous cellulose in moderate process conditions [1]. Deep eutectic solvents' promising properties have forced and encouraged their subsequent use for successful LCB bioconversion, making them superior to diluted acids and ionic liquids.

Jing et al. [80] investigated the production of biohydrogen from corn cob biomass by using a DES (ChCl/ethanolamine) system. The pretreatment resulted in maximum removal of lignin with high percentage efficiency (83.12%). Tan et al. [81] carried out pretreatment of oil palm empty fruit bunch (EFB) biomass with a choline chloride/formic acid (CC-FA), and choline chloride/lactic acid (CC-LA) solvent system. The results demonstrated improved biomass fractionation with high delignification efficiency of 60 wt%. Thi and Lee [82] investigated the effect of three DESs (choline chloride-urea, ChCl-U; chloride-lactic acid, ChCl-LA; choline chloride-glycerol, ChCl-G) on the pretreatment of oil palm empty fruit bunch biomass. The results illustrated that ChCl-LA pretreatment followed by saccharification gave the highest sugar yield (20.7%), followed by ChCl-G and ChCl-U pretreatment, which yielded sugar content of approximately 20.0% and 16.9%, respectively. To further improve the effectiveness of the DES pretreatment and maximize the yield of reducing sugars, a thorough investigation of the operating parameters and techno-economic analysis of the process is required.

### 4.2.3. Organosolv Pretreatment

Organosolv pretreatment is one of the promising methods for the effective valorization of lignocellulosic biomass into value-added chemicals that could facilitate the shift towards increased use of renewable biomass substrates. Organosolv pretreatment uses organic solvents (ethanol, methanol, formic acid, acetic acid, phenol, glycerol, or acetone) or their aqueous solutions to specially extract or remove lignin from lignocellulosic feedstocks [83]. The organosolv pretreatment provides for highly efficient generation of ethanol, lignin, and other biochemicals from LCB when compared to other pretreatment techniques. In this pretreatment, LCB is added to a combination of organic solvent and water that has a solid-to-liquid ratio between 1:4 and 1:10 (*w/w*) and a solvent concentration of 35–70% (*w/w*) [84]. The rate of pretreatment reaction can also be accelerated by including a catalyst. The pretreatment causes the lignin linkages and lignin–carbohydrate bonds to hydrolyze, resulting in a solid phase primarily consisting of cellulose and hemicellulose. To avoid lignin precipitation, this prepared material must be rinsed with an organic solvent. Thereafter, the pretreated biomass is washed in distilled water to remove the organic solvent, and the solid and liquid phases are separated. The pretreated solids are subsequently enzymatically saccharified and fermented to produce value-added biochemicals, whereas the solvent is recovered from the spent liquor using the distillation process (Figure 2).

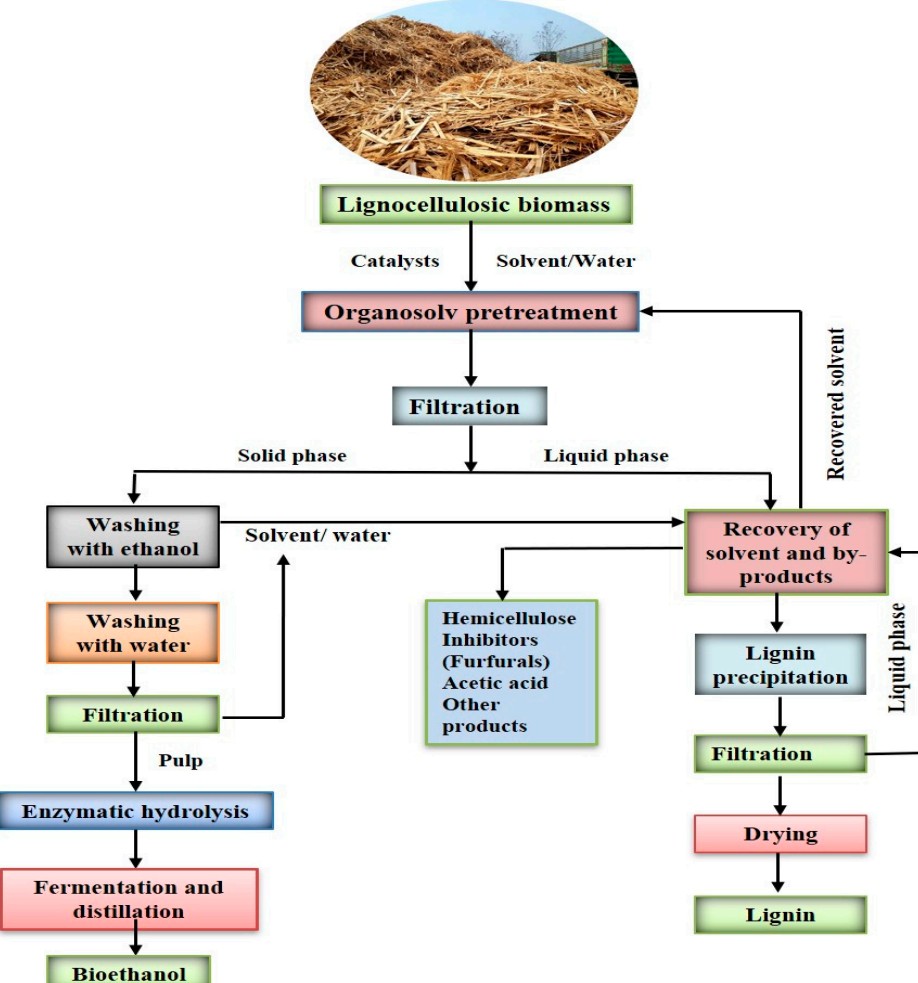

**Figure 2.** Schematic representation of steps in organosolv pretreatment of lignocellulosic feedstocks.

Karnaouri et al. [85] pretreated pine and beechwood biomass with mild oxidative organosolvs (tetrahydrofuran, acetone and ethanol) that produced cellulose-rich solid fractions for lactic acid production from *Lactobacillus delbrueckii* strain. A high lactic acid

production of 62 g/L was obtained from beechwood biomass compared to that obtained from pine biomass (36.4 g/L), showing that organosolv pretreatment is suited for both hardwood and softwood biomass for the production of valuable biochemicals. In another recent study [86], the sorghum biomass was pretreated by modified low-cost organosolv using glycerol solvent and ammonia as catalyst, which resulted in a significant improvement in the saccharification (reducing sugars of 421.35 mg/g biomass) and fermentation yields (36 g/L). Tang et al. [87] executed synergetic organosolv and organic amine pretreatment of corn stover biomass to produce salt-free high-quality lignin and fermentable sugars. Under optimal process conditions, maximum sugar yield of 83.2%, made up of 87.1% glucose and 75.4% xylose, was obtained with a significant delignification percentage of 81.7% by the application of aqueous ethanol (60% *v/v*) as a solvent, and *n*-propylamine (10mmoL/g biomass) as a catalyst.

The promising results with the organosolv pretreatment have led to the establishment of organosolv-based biorefineries in developed countries including the United states of America, Brazil, Germany, Australia, France, and Canada [84]. The further setting up of organosolv-based biorefineries would need an integration of LCB components into pretreatment processes and the reduction of pretreatment costs. This can be accomplished by lowering the volume of organic solvent used in pretreatment, raising the added value of by-products, and optimizing the entire process using statistical approaches.

*4.3. Physicochemical Pretreatment*

The physicochemical pretreatment technique involves the combination of both chemical and physical methods, and is believed to be more efficient in the disintegration of lignin–polysaccharide complexes under varying process conditions, i.e., temperature and pressure [88]. Various methods including carbon dioxide explosion, steam explosion, liquid hot water method, ammonia fiber explosion, wet oxidation, and different combinatorial approaches come under physicochemical methods [20,89]. Steam explosion is the most common method for LCB pretreatment in which an autohydrolysis reaction is carried out by high-pressure saturated steam, which decreases the biomass crystallinity, removes lignin, and transforms cellulose and hemicellulose into soluble oligomers [63]. In $CO_2$ explosion pretreatment, dissolution of LCB is caused due to the application of high-pressure supercritical carbon dioxide at high temperature. Ammonia fiber explosion and $CO_2$ explosion work at low temperature; thus, are considered more economical [90]. The integration of different techniques such as acids, alkali, DESs, ILs, combined with ultrasound, microwave, or other methods have been employed by various researchers to improve the effectiveness of the process [9,15,17].

Kuglarz et al. [91] carried out combined pretreatment of industrial hemp biomass with steam at 180 °C and 1% $H_2SO_4$, which resulted in the production of a maximum glucose yield of 73–74%, and an ethanol content of 75–79%. Nargotra et al. [35] reported an enhanced sugar yield of 63.42 mg/g biomass by the combinatorial pretreatment (ionic liquid, 1-butyl-3-methyl imidazolium chloride+ alkali, NaOH) of sunflower stalk biomass, as compared to IL (79.6 mg/g biomass), and NaOH (97.38 mg/g biomass) in standalone mode. Vaid et al. [92] reported a combined application of DES (choline chloride/glycerol) and calcium hydroxide for the pretreatment of *Saccharum spontaneum* biomass, followed by enzymatic hydrolysis under one-pot consolidated bioprocessing. The results demonstrated a maximum sugar yield of 372.3 mg/g biomass which was 4.94-fold higher than the sugar yield obtained under unoptimized conditions (75.25 mg/g biomass). In another study conducted by Hou et al. [93], eucalyptus sawdust was pretreated by combined application of microwave and IL [TBA][OH], which on hydrolysis yielded maximum sugar yield of 410.67 mg/g biomass in 48h under optimal pretreatment conditions. Vaid et al. [33] employed a novel pretreatment approach using sodium dodecylsulfate-assisted IL tris (2-hydroxyethyl) methyl- ammonium methyl sulphate for the deconstruction of *Saccharum spontaneum* biomass. Hydrolysis with IL-stable *Bacillus subtilis* G2 enzymes yielded

maximum reducing sugars of 364.24 mg/g biomass under optimized process conditions, in comparison to unoptimized (154.91 mg/g biomass).

### 4.4. Biological Pretreatment

Biological pretreatment entails the utilization of microorganisms, mostly fungi and bacteria, which have the ability to produce extracellular lignocellulolytic enzymes for the degradation of lignocellulosic biomass. Brown, white, and soft-rot fungi such as *Dichmitus squalens*, *Phanerochaete chrysosporium*, *Jungua separabilima*, *Phlebia radiata*, and *Rigidosporus lignosus* are generally used for the digestion of polysaccharides [20,34]. This pretreatment method is regarded as an efficient, affordable, and environmentally friendly process as it does not employ high energy inputs, and produces less harmful substances, compared to other chemical strategies [94]. However, less available surface area due to biomass structural complexity, slow digestion rate, and requirement of monitored microbial growth conditions restrict the use of biological pretreatment at commercial levels [63,95] (see Table 2).

**Table 2.** Different pretreatment methods with their merits, demerits, and pretreatment conditions and promising results.

| Pretreatment Process | Merit | Demerit | Biomass/Pretreatment Conditions | Significant Results | Reference |
|---|---|---|---|---|---|
| Acid treatment | Hemicellulose hydrolysis, increased biomass porosity | Formation of furfurals, hydroxymethyl furfural, corrosion | Sugarcane bagasse/4.95% phosphoric acid, 80 °C, 375 min) | 98% glucose yield, 99% saccharification efficiency. | [73] |
| | | | $\gamma$-Valerolactone/dilute $H_2SO_4$ (4:1, *v/v*), 120 °C, 60 min. | 89.1% glucose yield | [96] |
| Alkali pretreatment | High lignin removal, hemicellulose hydrolysis | Formation of magnesium and calcium salts, long residence time | Date palm/20 % $NH_3$, 80 °C, 12 h | High biochemical methane potential (309.76 mL $CH_4$/g-TS) | [74] |
| | | | Giant reed biomass/20% NaOH | High glucose yield (44.9%), high $H_2$ yield (98.3 mL/g TS) | [97] |
| Ionic liquid pretreatment | Liquid at room temperature, low toxicity, low vapor pressure, high digestibility, thermal stability | Expensive and toxic to hydrolytic enzymes | Almond wood/ethanolamine acetate (15 wt % solid loading) | High glucose (24–82%) and xylose yields (14–80%); 60.8% fermentation efficiency | [98] |
| | | | Stinging nettle stems/1-butyl-3-methylimidazolium acetate (10 g biomass in 50 $cm^3$ IL, 120 °C, 2 h) | High ethanol concentration (7.3 g $L^{-1}$) | [99] |
| Deep eutectic solvent pretreatment | Easy synthesis, low-cost, biodegradable less toxic, recyclable | High viscosity, hygroscopic | *Parthenium hysterophorus*/ChCl/sorbitol (1:5), | Higher sugar yield (148.54 mg/g biomass) | [16] |
| | | | Sugarcane bagasse/ChCl:glycerol (1:10)-ultrasound | Higher reducing sugar titer (276.8mg/g substrate) | [9] |
| | | | Banana peel waste/ChCl-based DES | High total reducing sugar yield of 72.9% | [100] |
| Organosolv pretreatment | Easy recovery and recycling, effective delignification | Repeated washings of pretreated LCB, expensive solvents | Pine, beechwood/mild oxidative (acetone, tetrahydrofuran, and ethanol) | High lactic acid production (beechwood: 62 g $L^{-1}$; pine: 36.4 g $L^{-1}$) | [85] |
| | | | Corn stover/aqueous ethanol (60%, *v/v*), n-propylamine (10 mmol/g, biomass) | High sugar yield (83.2%) and delignification (83.2%) | [87] |
| Ultrasound pretreatment | Size reduction, proper mixing of biomass with solvent, disintegration of cell wall components, less process time, assistive technique | Low conversion efficiency | Rice straw/ultrasound-IL treatment, | Increased reducing sugar, delignification and cellulose conversion by 20.13–28.96%, 18.06–19.33% and 31.69–35.23%, respectively | [101] |
| | | | Eucalyptus wood/ultrasound-distilled water (28 KHz, 300 W) | Effective disintegration of biomass with 35.5% increase in crystallinity | [64] |
| Microwave pretreatment | Continuous operation, less process time and low energy input | Requires high temperature for processing, no hot spots, low efficiency for apolar materials | *Brachiara mutica* (paragrass)/microwave-alkali (5% *w/w*), 120 °C, 30 min. | Increased total reducing sugars to 137.3% | [69] |
| | | | Rice straw/microwave radiation for 4 min, at 190 °C) | High specific yield of methane (325.23 mL/g/VS) | [70] |

## 5. Enzymatic Hydrolysis as an Integral Step in Biorefineries

Enzymatic hydrolysis of pretreated lignocellulosic feedstock is the key step in the LCB-biorefinery chain, which involves the transformation of polymeric carbohydrates into simple fermentable monomers with the help of saccharification enzymes such as cellulases, hemicellulases, and ligninases [14,33,55,61]. The three major cellulolytic enzymes employed to degrade cellulose are: a) endo-1,4-β-D-glucanases (hydrolyze low-crystalline cellulose chains randomly); b) cellobiohydrolases/exoglucanases (hydrolyze cello-oligosaccharides' reducing and non-reducing ends to generate cellobiose units); and c) cellobiases/β-glucosidases (hydrolyze cellobiose to form D-glucose) [2,4,36]. All the three cellulases work simultaneously and synergistically as a mixture, thus increasing the catalytic activity of cellulase enzymes [102].

Besides cellulases, hemicellulases such as β-xylosidase, xylanases, glucomannanase, glucuronidase, galactomannanase, and acetylesterase are employed for the hydrolysis of hemicellulose polymers [23,103,104]. Xylanases solubilize β-1,4-D-xylan fractions of hemicellulose by cleaving the β-1,4-D-xylopyranosyl bond, and thus, result in the formation of oligomeric (xylo-oligosaccharides) and monomeric sugars [105,106]. Another class of enzymes named lytic polysaccharide monooxygenases (LPMO) work synergistically with cellulases, and thus boost the effectiveness of cellulose degradation [4,23]. LPMOs are copper-dependent enzymes that follow an oxidative mechanism to improve the disintegration of biomass. In the presence of cellobiose dehydrogenase and ascorbic acid (electron donors), copper ions cause reduction of oxygen, thus aiding in the cleavage of glycosidic linkages of crystalline cellulose [36,58].

Lignin, an aromatic polymer of LCB, acts as a physical barrier in the process of polysaccharides extraction [1,4,15]. It is degraded by a variety of ligninolytic enzymes such as laccases, versatile peroxidase manganese peroxidase, and lignin peroxidase [107]. Micro-organisms (fungi and bacteria) are commonly used for the production of hydrolytic enzymes. Micro-organisms such as *Bacillus velezensis*, *Bacillus paranthracis*, *Aspergillus terreus*, *Aspergillus niger*, *Trametes versicolor*, and *Phanerochaete chrysosporium* are a few of the examples that produce enzymes with cellulolytic, hemicellulolytic, and ligninolytic capabilities [2,107]. Various factors such as enzymatic activity of saccharification enzymes, enzyme loading, substrate quantity, process conditions, and inhibitors present in the post-pretreatment sugar hydrolysate [90]. However, addition of various salts/surfactants/metal ions, optimization of process parameters, and utilization of enzyme mixture increase the efficiency of enzymatic hydrolysis [108].

In a study conducted by Cai et al. [109], the addition of polyvinylpyrrolidone (1 g/L) during enzymatic saccharification of dilute acid-pretreated eucalyptus biomass increased its digestibility to 73.4% from 28.9%, as compared to other additives (bovine serum albumin, PEG, and Tween). Rocha-Martin et al. [110] reported an improvement in the glucose yield by 32, 10, and 7.5% after PEG4000-assisted hydrolysis of pretreated microcrystalline cellulose (Avicel), corn stover, and sugar cane straw biomass, respectively. Addition of PEG4000 increased the endoglucanase and beta-glucosidase catalytic activity by 60 and 20%, respectively and also decreased the liquefaction time (up to 25%).

Although enzymatic hydrolysis is a crucial step in the bio-transformation of LCB at commercial level, application of enzymes still has a number of drawbacks, including high cost, limited availability, less stability, non-recyclability, and non-reusability; thus, limiting the effectiveness of the bioprocess [111]. To circumvent these problems, immobilization of microbial enzymes on suitable matrices/support materials provides a promising solution [112]. Nanomaterials (with a size of less than 100 nm) such as nanoparticles, nanofibers, nanoflowers, nanomagnets, and many others have the potential to completely transform the production of biocatalysts and their application in the bioenergy area [4]. Nanobiocatalysts provide a number of benefits to enzymes including high surface area, increased activity, increased stability, reusability, easy synthesis, environmental friendliness, and a cost-effective nature [113,114]. Furthermore, enzyme engineering or genetic manipulation of gene coding for saccharification enzymes can improve the catalytic activity and stability

of enzymes [4]. The production of competent and resistant microorganisms, which can survive harsh environments, and development of stable enzymes require the potential application of genetic engineering tools [25].

Karnaouri et al. [115] produced an enzyme called MtEG5A expressed from the endoglucanase gene of *Myceliophthora thermophile*, a thermophilic fungus, into *Pichia pastoris*, the methylotrophic yeast. These enzymes have been found to hydrolyze birch, wheat straw, and spruce biomass, producing large quantities of cellobiose and hence increased yield of sugars. In a study, cellulases immobilized on magnetic nanoparticles were utilized for the hydrolysis of *Allamanda schottii* L. flowers pretreated with sodium hydroxide, and the highest sugar yield of 25 g/ml was observed in comparison to that with native enzyme, i.e., 18 g/mL. Fermentation of the obtained sugar hydrolysates yielded an ethanol content of 252 g/L (immobilized enzyme), and 182g/L (native enzyme), respectively [116]. Hwangbo et al. [117] investigated the combined pretreatment of NaOH and IL, 1-ethyl-3-methylimidazolium chloride of corn husks, hydrolyzed with five types of saccharifying enzymes, which were cross-linked in the presence and absence of magnetic nanoparticles called magnetite-saccharifying enzyme cross-linked enzyme aggregates, M-SE-CLEAs, and saccharifying enzyme cross-linked enzyme aggregates, SE-CLEAs. The results indicated a maximum reducing sugar yield of 250 mg/g biomass by employing SE-CLEAs. Additionally, reusability of M-SE-CLEAs was successfully achieved in up to three hydrolysis cycles.

## 6. Bioconversion of Lignocellulosic Biomass into Biofuels and Value-Added Materials

Carbohydrate-rich lignocellulosic biomass obtained from agricultural by-products, forestry residues, municipal wastes, and industrial emissions holds a remarkable potential for the production of green energies, i.e., biofuels (bioethanol, biohydrogen, biomethane, biobutanol, biogas), and other value-added products through various aforementioned thermochemical and biochemical conversion technologies [1,59,79]. Enzymatic digestion of polysaccharides results in the production of reducing sugars, which on fermentation produce various kinds of biofuels. A biochemical process that catalyzes the conversion of monomeric carbohydrates with the help of microorganisms under a low oxygen environment is known as fermentation [26]. Most widely employed industrial microorganisms for the fermentation process are *Schizosaccharomyces*, *Saccharomyces cerevisiae*, *Candida shehatae*, *Zymomonas mobilis*, *Pichia stipitis,* and others [90]. For microorganisms to thrive, all the internal conditions (dissolved oxygen, culture medium, and other micronutrients), and necessary parameters (pH, temperature, and sugar content) must be satisfied. They must possess high product yield, and resistance to the toxic inhibitory compounds [118]. The conventional method used for fermentation of biomass hydrolysate has a specific process in which the fermentation is carried out in several types (batch, fed-batch, solid state, consolidated bioprocessing, simultaneous saccharification and fermentation, and separate hydrolysis and fermentation) [23,27]. The processed lignocellulosic biomass after several rounds of steps, including pretreatment and enzymatic saccharification, acts as a suitable substrate for the production of bioproducts such as biofuels, bioplastics, vanillin, hydroxymethylfurfural, etc. (Figure 3) [1,4,103,119].

### 6.1. Bioethanol

Biofuel ethanol ($C_2H_5OH$) is a promising green combustible fuel compared to gasoline due to various characteristics such as higher flammability, high octane number, higher combustion efficiency, high heat of vaporization, shortened burning time, and minimum toxicity, and is less polluting than petrol. It is usually mixed with gasoline and used as fuel in the transportation industry [27,120]. India contributes only 2% to the overall global bioethanol production [121]. At present, most of the ethanol is produced from first-generation (1G) feedstocks, which are a proven and well-established method with high bioethanol productivity and output. Data from the International Energy Agency (IEA) show that 1G bioethanol production has increased globally to 104 billion liters with

a 3% yearly growth rate [122]. However, the process is linked to the controversy over the conversion of food into fuel and leads to significant environmental impacts [123]. In contrast, second-generation (2G) bioethanol production is built on the limitations of 1G bioethanol and is mostly generated from lignocellulosic feedstocks in the form of non-food crops and agricultural wastes. According to IEA reports, a 2% annual growth rate for bioethanol production is estimated, which will account for 119 billion liters of bioethanol in 2023 and might increase to 145 billion liters under favorable circumstances [122]. For the production of bioethanol, LCB is first processed into simple sugars (hexoses and pentoses), and then metabolized into ethanol by fermenting microorganisms such *Saccharomyces cerevisiae* and *Pichia stipitis* [2]. Jin et al. [124] executed ethanol fermentation of the sugary liquor obtained by hydrolysis of combinatorially (sodium hydroxide and hydroxymethylation reagents) pretreated sugarcane bagasse, and observed a maximum ethanol yield of 10.67 g/L. Vaid et al. [17] reported the highest ethanol concentration of 0.148 g/g pine needle biomass under consolidated bioprocessing of pine needle biomass pretreated with 1-ethyl-3-methylimidazolium methanesulfonate. However, despite remarkable research being done on the bioethanol production from LCB using novel biorefinery approaches, the cost of production has been a major bottleneck in the commercialization of this green fuel. Therefore, some of the limitations associated with standalone 1G and 2G processes may be resolved by integrating 1G–2G bioethanol production, which could ultimately result in a breakthrough in the creation of a workable technology for large-scale bioethanol production [123]. When compared to a standalone 2G bioethanol plant, integrated 1G–2G bioethanol production from sugar-based crops offers better productivity rates, an appealing commercial opportunity, and is more ecologically benign.

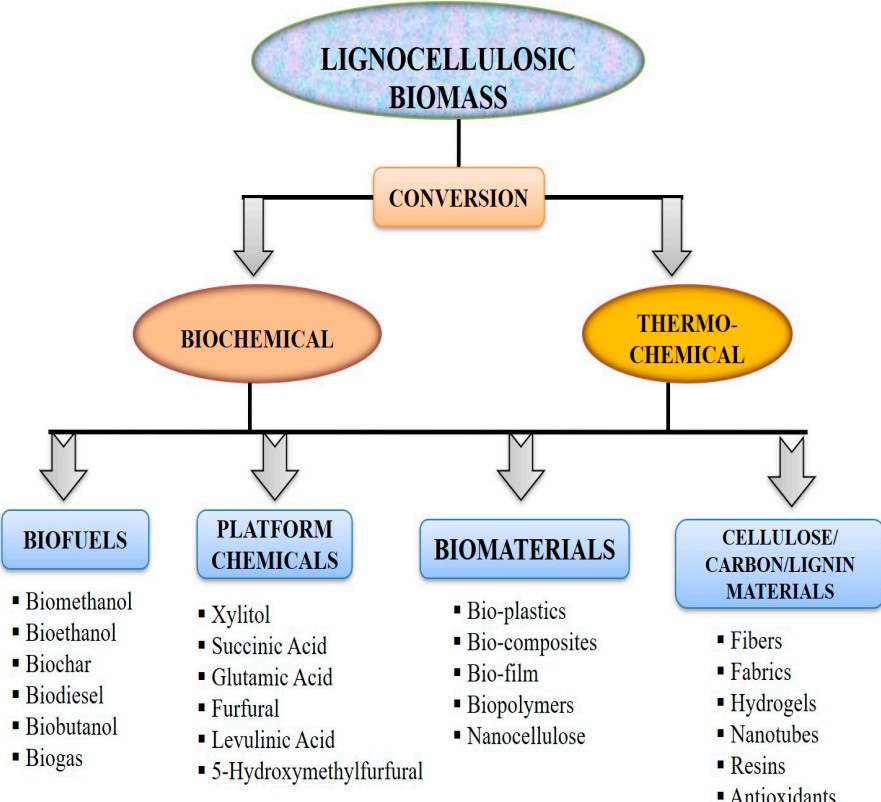

**Figure 3.** Biochemical and thermochemical conversion of lignocellulosic substrate into biofuels and value-added bioproducts.

*6.2. Biodiesel*

Another promising alternative to non-renewable fossil fuels is fatty acid methyl ester, called biodiesel due to its sustainability, biodegradability, renewability, and environmental

friendliness as compared to conventional diesel [125]. It is generally produced from edible and non-edible oils by transesterification, blending, pyrolysis, microemulsion, or other conversion processes. Biodiesel can replace the fossil-derived diesel directly without any modification in existing engines, and is non-explosive, non-inflammable, and non-toxic [126]. Wang et al. [127] reported two insect (*Tenebrio molitor* L. and *Hermetia illucens* L.)-based biorefinery processes of corn stover biomass, and observed the produced crude grease of 1.95 g yielded a significant quantity of biofertilizer (111.59 g), protein (6.55 g), and biodiesel (1.7g). Bateni et al. [128] carried out alkali pretreatment of castor plant at optimal conditions to improve the bioethanol and biodiesel yield, which was obtained to be 27.2–71% and 85.0 ± 1.0%, respectively. It was estimated that 149.6 g of biodiesel and approximately 30.1 g of bioethanol was produced from 1 kg of castor plant.

### 6.3. Biomethane

LCB biopolymers can be digested anaerobically by bacteria to produce an easily storable and flexible biofuel named biomethane or biogas [12]. Firstly, the complex ligno-polysaccharides are processed with a suitable pretreatment strategy into simpler molecules of amino acids, sugars, and fatty acids. Acidogens (fermentative bacteria) digest these monomers to short chain fatty acids, and then convert them into acetate, hydrogen, and carbon dioxide. These end products act as substrates to produce biomethane via methanogenesis. Recently, Banu et al. [129] reported a maximum biomethane yield of 0.174 L/g COD from waste-activated sludge pretreated by bacterial pretreatment induced by mild hydrogen peroxide, as compared to methane obtained with bacterial pretreatment, i.e., 0.078 L/g COD, and control sludge, i.e., 0.02175 L/g COD.

### 6.4. 5-Hydroxymethylfurfural (HMF)

5-Hydroxymethylfurfural (HMF) is generally considered as a multifunctional molecule due to the presence of a furan ring structure joined with an alcohol and aromatic aldehyde [130]. 5-Hydroxymethylfurfural can produce a variety of derivatives including 2,5-diformylfuran (2,5-DFF), levulinic acid, 2,5-dimethylfuran (2,5-DMF), and 2,5-bis(hydroxymethyl) furan (2,5-BHF) [131]. These compounds have been used as precursors in the production of materials such aspolyesters, polyamides, and polyurethane [132]. Ruby and Schüth [108,133] reported 5-hydroxymethylfurfural production by fructose dehydration with the application of acid-free and metal-free heterogeneous catalytic systems. The results showed a maximum of 77% HMF yield. Zhao et al. [134] carried out as-synthesized solid acid catalyst-mediated fructose degradation to produce HMF with maximum yield of 90% under optimal process conditions.

### 6.5. Biohydrogen

Hydrogen is a clean fuel, which when burned in fuel cell releases water and possesses a greater combustion rate. In the recent past, several biohydrogen production methods have been investigated from a variety of lignocellulosic materials, including rice straw, wheat straw, grass, softwood, rice husk, and woody waste. The $H_2$ production from LCB involves a step-wise sequence of pretreatment, hydrolysis, fermentation, and gasification. The biological approach, i.e., dark fermentation, is thought to be more environmentally benign and long-lasting when compared to other modern technologies [135]. Using locally isolated facultative bacteria, *Proteus mirabilis*, the hydrogen production from the hydrolysates of acid and alkali pretreated paddy straw was explored. A maximum volume of 833.43 ± 21.72 mL $H_2$ was produced from acidic hydrolysate through dark fermentation, which had a higher $H_2$ yield than alkaline treatment method [136]. Utilizing corn silage, fruit and vegetable waste, and sugar beet pulp, dark fermentative hydrogen production was studied that produced 52 $cm^3$/gVS $H_2$ after deep eutectic solvent pretreatment of waste biomass [137]. In another study, the microwave pretreatment of switchgrass and miscanthus at various processing temperatures and dissolution in subcritical water resulted in a higher yield of hydrogen-rich gas by aqueous-phase reforming [138]. However, more

innovative strategies are required to improve the viability of lignocellulosic biohydrogen production, and integrated biorefinery platforms that may offer a viable alternative.

## 7. Environmental Sustainability of Biofuels and Value-Added Biochemicals from Lignocellulosics

The continuous emission of greenhouse gases (GHGs) due to combustion of fossil-derived fuels lead to an enormous increase in the global temperature, causing global warming, and other environmental problems including pollution, climate change, and oil spills [27]. An exponential rise in global population and depletion of conventional fossil resources have caused the demand for energy worldwide to soar [139]. Valorization of abundantly available lignocellulosic biomass waste into biofuels and value-added products promises an effective solution. Biofuels, which are made from non-fossilized biomass (plant products) or other organic solid wastes, offer the advantage of lowering net carbon emissions, primarily GHGs, and dependency on oil [121]. Increasing the use of biofuels would benefit several important policy goals such as energy security, fuel quality, and environmental friendliness. Biofuels can easily replace petroleum-based fuels and, in many nations, can serve as an indigenous source of transportation fuel rather than one that must be imported. However, if imported, biodiesel or ethanol will probably originate outside of OPEC (Organization of Petroleum Exporting Countries), diversifying the supply of transportation fuels on a more global scale.

The advantages of using ethanol to increase fuel octane levels have caught the attention of refiners and automakers, particularly in cases where MTBE and other possible octane enhancers are restricted or banned. Biofuels are often more environmentally benign than petroleum fuels due to their production of fewer GHG emissions across the whole fuel cycle. Vehicles powered by biofuels release less of various pollutants that exacerbate air quality issues, especially in metropolitan areas, either in their pure form or as blends with conventional petroleum fuels. Combining biofuels with conventional fuels can reduce some air pollutants while increasing others (such as NOx emissions) [140]. The bioconversion of LCB into various value-added chemicals such as formic acid, succinic acid, lactic acid, levulinic acid, 2,5-dimethylfuran (2,5-DMF), 5-hydroxymethylfurfural (5-HMF), and other aromatic compounds can be transformed into a variety of upgraded derivatives that possess remarkable potential as biopolymers, biofuels, and in solvent factories [131].

## 8. Current Challenges and Future Prospects

Second-generation biorefinery based on lignocellulosic feedstock is a renewable, sustainable, and eco-benign approach in view of deteriorating fossil fuel supplies, increased energy prices, and environmental ill-effects [59]. However, full-scale bio-transformation of LCB into desired chemicals is still limited due to high operational and investment costs. This challenge is ascribed to LCBs' intrinsically tough structure, which resists microbial invasion and enzyme access [13]. A comprehensive study on the characterization and composition of the lignocellulosic substrate is a prerequisite for the development of a successful biorefinery. The composition, availability, cost, production, collection, storage, and shipping are factors which are taken into consideration while selecting lignocellulosic feedstock [2]. Bioconversion processes including biomass pretreatment and enzymatic saccharification are the key steps that affect the techno-economic feasibility of a biorefinery. As the green refinery concept is put into practice, it is necessary to maximize the cost-effectiveness and quality of fractionated products. Despite improvements in a number of different pretreatment strategies, which produce high sugar yield with fewer demerits, there is still a lot of room for enhanced LCB pretreatment and detoxification-based advancements [59]. Thus, adequate and cost-efficient pretreatment methods need to be developed that can cause maximum degradation of biomass, with less chemical and energy consumption, and boost the subsequent enzymatic hydrolysis. The severity of biomass pretreatment influences the amount of enzyme required for the next step of hydrolysis [27]. Therefore, future research efforts should focus more on developing customized pretreatment techniques while con-

sidering various LCB compositions or the targeted products, as well as developing a solid understanding of the reaction mechanism underlying pretreatment.

Another major obstacle that hinders the feasibility of a biorefinery is the high operational expense of hydrolytic enzymes. The enzymatic hydrolysis step contributes about 25–30% of the total biomass processing cost, and has not yet been fully commercialized for second-generation biofuel production. Therefore, exploration and production of affordable enzyme mixtures are required for an efficient hydrolysis process [90]. Several techniques such as metagenomics, metabolic engineering, and protein engineering can be used to discover highly stable and genetically modified hydrolytic enzymes, leading to the production of various desirable products [4,25]. Immobilization of potent enzymes on nanosized materials (organic wastes, nanoparticles, nanofibers, nanotubes, and others) provides a plethora of benefits to enzymes including high stability, recyclability, reusability, and high catalytic activity, thus improving the cost effectiveness of 2G biorefinery [117]. The majority of the microorganisms involved in fermentation can metabolize only a specific class of sugars; for example, *Saccharomyces cerevisiae* can ferment only hexoses, while *Pichia stipitis* can metabolize only five carbon sugars, leaving other residues unused (lignin and unreacted substrates, among others). To boost ethanol yield and productivity, it is crucial to investigate and create genetically modified cellulolytic and fermentative microorganisms and co-culture systems. Future commercialization of biorefineries is just the tip of the iceberg. Due to the high cost of renewable energy produced by a single technology, combining various technologies on a large scale for commercial purposes would increase the yield of renewable energy and make it more economical. The integration of biorefineries to scale up the projects to close the gap between the generation and commercialization of biofuels and value-added products is the subject of extensive research. In the past decade, a number of research and review articles have been published in the literature that have provided exciting and novel low-cost methods of the valorization of lignocellulosic biomass into biological commodities including biofuels (Figure 4). An analysis of the articles published from 2013 to 2022 shows a total of 927 research and review papers (through a Scopus database search for the terms 'biomass AND lignocellulose', 'biofuels' and 'pretreatment' in the abstracts, titles, and keywords).

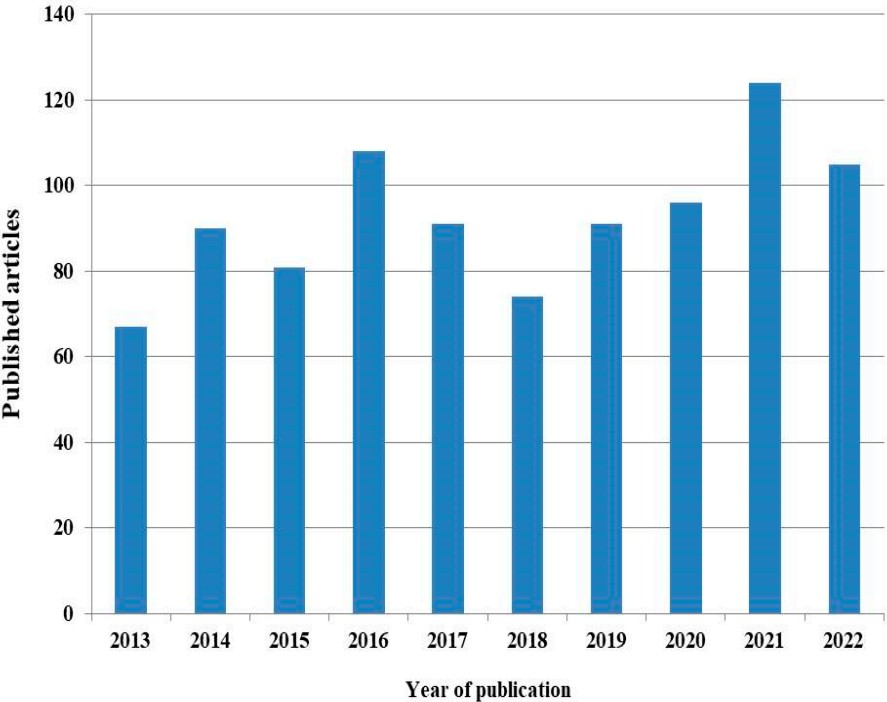

**Figure 4.** Number of articles published between 2013 and 2022 (Scopus database: lignocellulosic biomass/biofuel/pretreatment in the title/abstract/keyword of a research or review paper).

Over the past ten years, there has been a lot of interest in using LCB to produce fuels and value-added products, and therefore, a number of international patents have also been obtained, both at pilot and industrial scales. Patents provide both economic and technological information, and show the general tendency of markets. The United States, China, Japan, Germany, South Korea, France, Canada, and the United Kingdom have filed the highest number of patents related to biofuels [141].Different processes, techniques, and products have been developed and patented from time to time. Future research can be carried out on patent development technology with a focus on economic sustainability. In the coming years, the extensive and fruitful research on the bioconversion of LCB into 2G biofuels and value-added materials is expected to rise, ideally fulfilling the objectives of the lignocellulosic biorefineries.

## 9. Conclusions

The concept of lignocellulosic biorefinery is developed by valorizing lignocellulosic biomass and its constituent parts into value-added products and enhanced energy. Introduction of process-efficient and low-cost technologies is the most essential requirement for sustainable and renewable energy production. It not only minimizes the negative environmental impact, but also cuts down on carbon emissions. This review encompasses detailed information on the lignocellulosic biomass, processing technologies, merits, demerits, and challenges associated with a biorefinery. For forest-based biorefineries to be incorporated into industrial supply chains, a sustainable and ecologically friendly solution for the valorization of lignocellulosic biomass through cost-effective and sustainable processes must be developed.

**Author Contributions:** Conceptualization, V.S. and P.N.; validation, V.S., M.-L.T., C.-W.C., P.N. and C.-D.D.; formal analysis, V.S., B.K.B., M.-L.T., P.N. and C.-W.C.; investigation, V.S., B.K.B. and C.-D.D.; writing—original draft preparation, S.S. and V.S.; writing—review and editing, P.N. and V.S.; visualization, V.S., M.-L.T. and C.-D.D.; resources, M.-L.T.; supervision, M.-L.T., P.-P.S. and C.-D.D.; funding acquisition, C.-D.D. All authors have read and agreed to the published version of the manuscript.

**Funding:** This work was funded by Taiwan MOST for funding support (Ref. No. 109-2222-E-992-002).

**Data Availability Statement:** Not applicable.

**Acknowledgments:** Authors V.S. and P.N. gratefully acknowledge National Kaohsiung University of Science and Technology, Kaohsiung City, Taiwan, for providing the post-doctoral fellowship. Authors M.-L.T., C.-W.C. and C.-D.D. acknowledge Ministry of Science and Technology, Taiwan MOST for funding support (Ref. No. 109-2222-E-992-002).

**Conflicts of Interest:** The authors declare no conflict of interest.

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
