# Peer review of "Environment Friendly Pretreatment Approaches for the Bioconversion of Lignocellulosic Biomass into Biofuels and Value-Added Products"

_environments, doi:10.3390/environments10010006_

Round 1

Reviewer 1 Report

The Review Manuscript ID environments-2054606, entitled “Valorization of lignocellulosic biomass into biofuels and value-added products for sustainable environment: A comprehensive review” deals with a hot topic reviewed in the literature. The following recommendations are made:  

Major aspects:

It is highly recommendable to include at the end of the Introduction a paragraph to clearly specify in which aspects does the present literature review complement other related reviews already published in the literature over the years. It is also recommendable to include a Table (which may be added as Supplementary material) listing the publication year and title of all related reviews (also indicating the reference of each review) within a period of 5-10 years. 

A literature search for review articles by “biomass and lignocellulos* and biolog* and biofuel and (valori* or value)” gives 113 results from Web of Science Core Collection. The corresponding publications record may be found in https://www.webofscience.com/wos/woscc/citation-report/648a1c10-da31-4e27-98ce-0df6f4995f6b-6398b908

Possibly a plot of publications of reviews per year in this topic may be included.

The classification of Fig 2 is confusing in that the physical treatments sometimes involve chemical reactions, as specified for example in section 4.1.1. Physical treatments should not involve chemical reactions, although they may involve the use of chemicals as solvents. Chemical treatments involve chemical reactions and may be classified as chemical treatments carried out using conventional energy supply methods (conduction/convection heat transfer, mechanical agitation) and, on the other hand, chemical treatments carried out using unconventional energy supply methods (ultrasounds, microwaves). The classification and reorganization of the information regarding Figure 2 is important for the sake of clarity and rigor.

Section 8 has many comments regarding economic issues, but the review does not present this perspective analysis.

It would be interesting to list some related and relevant patents.

Minor aspects:

“residence duration” should read “residence time”.

“100W” should read “100 W”.

Table 1, “Hemicellulose” should read “Hemicelluloses”.

If possible, indicate a wt% percentage range for “little amount of extractives”.

The values of wt % indicated throughout the text (e.g., page 6) should specify if they are on a dry mass basis or a different mass basis.

Physicochemical is sometimes spelled wrongly or with dash. Please correct this along the manuscript.

The following is confusing, “solid-to-liquid ratio of between 1:4 and 1:10 (w/w) and a solvent concentration of 35%–70% (w/w)”. Please specify it the liquid is the same as the solvent or what is the difference?

Author Response

Reviewers comments and responses:  

Reviewer 1: Comments

Comments and Suggestions for Authors

The Review Manuscript ID environments-2054606, entitled “Valorization of lignocellulosic biomass into biofuels and value-added products for sustainable environment: A comprehensive review” deals with a hot topic reviewed in the literature. The following recommendations are made: 

Major aspects:

Comment: It is highly recommendable to include at the end of the Introduction a   paragraph to clearly specify in which aspects does the present literature review complement other related reviews already published in the literature over the years. It is also recommendable to include a Table (which may be added as Supplementary material) listing the publication year and title of all related reviews (also indicating the reference of each review) within a period of 5-10 years.

Response: Thank you so much for in-depth investigation of our manuscript and your valuable comments which will motivate us to write even better for our future manuscripts. As per your suggestion, we have added a brief paragraph at the end of introduction highlighting the major aspects of the present review in the revised Manuscript (MS). Additionally, we have now added a new table listing the publication years and title of all related reviews within last 10 years.

Comment: A literature search for review articles by “biomass and lignocellulos* and biolog* and biofuel and (valori* or value)” gives 113 results from Web of Science Core Collection. The corresponding publications record may be found in https://www.webofscience.com/wos/woscc/citation-report/648a1c10-da31-4e27-98ce-0df6f4995f6b-6398b908. Possibly a plot of publications of reviews per year in this topic may be included.

Response: As per your kind suggestion, we have now added a new figure and a small paragraph in section 8 (Current challenges and future prospects) showing a plot of both research and review publications from last ten years (2013-2022) in the revised MS.

Comment: The classification of Fig 2 is confusing in that the physical treatments sometimes involve chemical reactions, as specified for example in section 4.1.1. Physical treatments should not involve chemical reactions, although they may involve the use of chemicals as solvents. Chemical treatments involve chemical reactions and may be classified as chemical treatments carried out using conventional energy supply methods (conduction/convection heat transfer, mechanical agitation) and, on the other hand, chemical treatments carried out using unconventional energy supply methods (ultrasounds, microwaves). The classification and reorganization of the information regarding Figure 2 is important for the sake of clarity and rigor.

Response: Thank you again for your valuable inputs regarding each and every section of the present MS. Respected Sir/Madam, the figure 2 has been added in the MS to provide a quick and brief idea to the reader regarding the different types of pretreatment methods such as physical, chemical, physicochemical and biological. We have thoroughly gone through the literature, citing previous published papers review/research) in the MS, and found that pretreatments such as ultrasound and microwave are categorized as ‘physical pretreatment’ methods. In the present figure 2, we have categorized the pretreatments on the basis of the kind of effect they produce on the biomass. Every kind of pretreatment method, regardless of them being physical/chemical/biological, eventually involves the breakdown of the complex linkages in the lignocellulosic biomass involving chemical breakdown. The ultrasound/microwave techniques first cause the mechanical breakdown of the recalcitrant biomass, and then involve the biochemical breakdown of the linkages between cellulose, hemicellulose and lignin fractions. Similarly, chemical pretreatment methods are the chemicals (organic/inorganic) that deposit on the biomass surface causing chemical reactions.

However, as per your suggestion, in figure 2 now we have added word ‘Mechanical’ along with physical pretreatment in the revised MS to avoid any confusion to the reader.  

Comment: Section 8 has many comments regarding economic issues, but the review does not present this perspective analysis. It would be interesting to list some related and relevant patents.

Response:  The information about the patents related to biofuels and value-added products from lignocellulosic biomass has now been provided in section 8 of the revised MS. 

 Minor aspects:

Comment: “residence duration” should read “residence time”.

Response: As per your suggestion the word ‘residence duration’ has been replaced by ‘residence time’ in the revised manuscript.

Comment: “100W” should read “100 W”.

Response: A space has now been provided at the required place to read it as ‘100 W’.

Comment: Table 1, “Hemicellulose” should read “Hemicelluloses”.

Response: The desired change has now been made in the revised MS.

Comment: If possible, indicate a wt% percentage range for “little amount of extractives”.

Response:   As per your suggestion we have now added wt% for all the components mentions in the text, including the extractives in the revised MS.

Comment: The values of wt % indicated throughout the text (e.g., page 6) should specify if they are on a dry mass basis or a different mass basis.

Response:  We have now specified the wt% of the different components of lignocellulosic biomass in dry mass basis in the revised MS.

Comment: Physicochemical is sometimes spelled wrongly or with dash. Please correct this along the manuscript.

Response: The pointed error ‘physico-chemical’ has now been corrected and changed to ‘physicochemical’ uniformly throughout the revised MS.

Comment: The following is confusing, “solid-to-liquid ratio of between 1:4 and 1:10 (w/w) and a solvent concentration of 35%–70% (w/w)”. Please specify it the liquid is the same as the solvent or what is the difference?

Response: The typological error has now been corrected in the revised MS. The statement has been said according to a published paper (https://doi.org/10.1063/1.5025876) and means that in the organosolv pretreatment, the organic solvent and water are mixed to give a solvent concentration of 35%–70% (w/w) and added to the biomass with a ratio of dry biomass to the solvent/water mixture (the solid:liquid ratio) between 1:4 and 1:10 (w/w).

Reviewer 2 Report

Dear authors, please find attached my comments to the manuscript

Author Response

Reviewers comments and responses:  

Reviewer 2: Comments

The present work reviews the pretreatment strategies for the conversion of lignocellulosic biomass into valuable products, both under an energetic and material point of view. Firstly, the biorefinery concept is presented, and linked to the circular economy approach; then the target feedstock is described showing its main constituents. A big chapter is than devoted to the description of pretreatment technologies, followed by the enzymatic hydrolysis and the conversion to fuels and chemicals. The paper has a good quality and fits with the scope of the Journal. It is well structured and clearly written, with adequate references to provide the state of the art. In my opinion it can be published after that the following points will be addressed

Comment: The title should be modified putting more emphasis on the pretreatment stage, since this is the main core of the review, while at the moment it seems more focused on the conversion step

Response: According to your valuable suggestion, we have now revised the title of the review manuscript with more emphasis on the pretreatment stage.

Comment: Recent reviews on the topic should be cited, also highlighting the difference of the present work and its novelty.

Response: According to your valid inputs on the review MS, we have now added some new references from last 2-3 years at various places of the MS and have also highlighted the novelty of the present MS in the introduction section. 

Comment: A table is suggested at the end of chapter 4 to summarize key findings, peculiarities, strength and weak points etc. regarding the different pretreatment technologies

Response:  A suitable table (Table 2) at the end of chapter 4 has now been added to the revised MS to summarize key findings, peculiarities, strength/weak points of different pretreatment technologies.

Comment: In paragraph 6.1 more information should be given due to the importance of the topic, and being bioethanol the most important biofuel in terms of annual production. For example, explain the important differences between 1st and 2nd generation bioethanol, and the complexity of the latter (some of these aspects are cited in chapter 8, but should be reported here as well)

Response: The section 6.1 has now been revised and as per your valuable suggestion, we have added a paragraph on 1st and 2nd generation bioethanol production in the revised MS.

Comment: In my opinion, chapter 6 should also contain a paragraph on hydrogen production from biomass, since it is an important topic being a strategic green energy vector. For example, aqueous phase reforming is a process which can start from lignocellulosic biomass and produces hydrogen (see for example the review of Zoppi et al. 10.1016/j.cattod.2021.06.002). I suggest using references from the works of Pipitone et al., Oliveira et al., and Irmak et al. to have suitable references for this paragraph.

Response: As per your valuable suggestion, we have now introduced a separate subsection as 6.5 on Biohydrogen production from lignocellulosic biomass in the revised MS to bring more value and strength to Chapter 6.

Comment: A grammar check should be performed throughout the manuscript to remove mistakes and typos

Response: Your valuable suggestion has been taken care of and we have checked the whole manuscript now to correct some of the grammatical mistakes end English language to make the sentences into meaningful form in the revised MS.

Round 2

Reviewer 1 Report

The authors have improved their work meriting publication